# RNA Editing Enzyme ADAR1 Regulates METTL3 in an Editing Dependent Manner to Promote Breast Cancer Progression via METTL3/ARHGAP5/YTHDF1 Axis

**DOI:** 10.3390/ijms23179656

**Published:** 2022-08-25

**Authors:** Yi Li, Ning-Xi Wang, Chuan Yin, Shan-Shan Jiang, Jia-Chu Li, Sheng-Yong Yang

**Affiliations:** 1Department of Biochemistry and Molecular Biology, Molecular Medicine and Cancer Research Center, College of Basic Medicine, Chongqing Medical University, Chongqing 400016, China; 2Department of Oncology, The First Affiliated Hospital of Chongqing Medical University, Chongqing 400016, China

**Keywords:** ADAR1, METTL3, breast cancer, ARHGAP5, YTHDF1

## Abstract

A-to-I RNA editing and m^6^A modification are two of the most prevalent types of RNA modifications controlling gene expression in mammals and play very important roles in tumorigenesis and tumor progression. However, the functional roles and correlations of these two RNA modifications remain to be further investigated in cancer. Herein, we show that ADAR1, an A-to-I RNA-editing enzyme, interacts with METTL3 and increases its protein level to promote the proliferation, migration and invasion of breast cancer cells through a mechanism connecting ADAR1, METTL3 and YTHDF1. We show that both ADAR1 and METTL3 are upregulated in breast cancer samples, and ADAR1 positively correlates with METTL3; ADAR1 edits METTL3 mRNA and changes its binding site to miR532-5p, leading to increased METTL3 protein, which further targets ARHGAP5, recognized by YTHDF1. Additionally, we show that loss of ADAR1 significantly inhibits breast cancer growth in vivo. Collectively, our findings identify the ADAR1–METTL3 axis as a novel, important pathway that connects A-to-I editing and m^6^A RNA modifications during breast cancer progression.

## 1. Introduction

Base modifications in mRNA, such as adenosine to inosine deamination (A-to-I editing), N6-methyladenosine (m^6^A) modification and cytosine to uracil deamination, play pivotal roles in various types of biological and pathological processes [1]. A-to-I editing and m^6^A modification are two of the most common RNA modifications controlling gene expression in mammals, and they have emerged as hot topics in cancer research [2,3]. A-to-I RNA editing is carried out by the adenosine deaminases acting on RNAs (ADARs: ADAR1/2) through conversion of adenosine to inosine within cellular RNA in mammals [4]. A-to-I editing mediated by ADAR1 primarily determines mRNA fate by affecting the code and stability of mRNAs, leading to changes in amino acid sequences, splicing, microRNA targeting and microRNA maturation [4]. Moreover, ADAR1 has two types of isoforms: ADAR1-p150 and ADAR1-p110. ADAR1-p150 is interferon (IFN)-inducible and is located in both the cytoplasm and nucleus [5,6], while ADAR1-p110 is constitutively expressed in the nucleus. Studies have shown that ADAR1 expression is elevated in breast cancer [7,8,9,10,11] and A-to-I editing mediated by ADAR1 changes its targets’ levels in breast cancer [12,13,14,15]. We have previously shown that ADAR1 is increased in breast cancer samples and promotes the proliferation, migration and invasion of breast cancer cells [16]. However, the mechanism underlying this is still to be explored.

m^6^A, another important RNA modification occurring mainly in the consensus sequence RRACH [17,18], is catalyzed by the m^6^A methyltransferase complex (writers) containing the core dimer of methyltransferase-like 3 and 14 (METTL3 and METTL14) [19,20,21]. The demethylation of m^6^A is carried out by FTO and ALKBH5 demethylases (erasers). In addition, m^6^A-binding proteins (readers), including YTH family proteins (YTHDF1, YTHDF2, YTHDF3, YTHDC2 and YTHDC1), specifically recognize m^6^A in order to affect the various aspects of RNA metabolism, such as translation, transport, localization, stability and alternative splicing. Dysregulation of m^6^A writers, erasers and readers is strongly associated with human diseases, especially cancers [22,23,24]. Dysregulation of METTL3 and METTL14 has been found in liver cancer [25,26], lung cancer [23] and glioblastoma [27]. Furthermore, FTO was reported to be an oncogene in AML [28] and lung squamous cell carcinoma [29]. However, the status of m6A and its correlation with ADAR1 in breast cancer still remain largely unknown.

In this study, we investigated the direct relationship between ADAR1 and METTL3 and found that overexpression and knockdown of ADAR1 respectively increased and decreased the expression levels of METTL3 and ARHGAP5 protein in breast cancer. We demonstrate that ADAR1 promotes the proliferation, migration and invasion of breast cancer cells through the METTL3/ARHGAP5/YTHDF1 axis and establish that METTL3 is one of the main targets of ADAR1 controlling biological functions.

## 2. Results

### 2.1. ADAR1 and METTL3 Proteins Are Upregulated in Both Breast Cancer Tissues and Cell Lines

To examine endogenous protein expression levels of ADAR1 and METTL3 in human breast cancer tissues and cells, immunohistochemistry (IHC) and Western blotting were performed. We found that both ADAR1 and METTL3 protein expression levels were increased in breast cancer tissues compared to normal adjacent tissues (Figure 1a,d) and in breast cancer cell lines compared to human breast epithelial cell MCF-10A (Figure 1b,e). Public datasets were used to determine the prognostic potentials of ADAR1 and METTL3 in breast cancer. Kaplan–Meier analysis indicated that higher ADAR1 and METTL3 expression levels are correlated with shorter overall survival (Figure 1c,f), which implies that both ADAR1 and METTL3 could promote human tumorigenesis and cancer development. Further, we found that ADAR1 expression was positively correlated with METTL3 expression in breast cancer samples in three different datasets (Figure 1g–i). Considering the above findings, we investigated whether ADAR1 regulates METTL3 expression to promote breast cancer development and progression in breast cancer cells.

### 2.2. ADAR1 Upregulates METTL3 and Increases mRNA m^6^A Levels in Breast Cancer Cell Lines

Previous studies have shown that ADAR1 and METTL3 promote the occurrence and development of tumors [16,25,30], respectively. Importantly, ADAR1 and METTL3 are two types of the most popular RNA modification events affecting adenosines, which motivated us to explore whether ADAR1 plays a role in breast cancer progression by regulating METTL3. First, we overexpressed ADAR1 protein in breast cancer cell lines. METTL3 protein and mRNA levels were found to be significantly increased, as shown in Figure 2a–d, from Western blotting and quantitative real-time PCR (qRT-PCR), while the opposite results were observed for the loss of ADAR1 expression in breast cancer cell lines (Figure 2e–h). Second, as METTL3 is a kind of methyltransferase that catalyzes the methyl group to the nucleotide of RNA, especially mRNA, we decided to determine whether ADAR1 could affect methylation of mRNAs by using the m^6^A dot blot assay in breast cancer cell lines, as shown in Figure 2i–k. mRNA m^6^A level was increased after ADAR1 overexpression in breast cancer cell lines, while the opposite effect was obtained in ADAR1 loss cells. Further, knockdown of METTL3 could decrease the heightened mRNA m^6^A level caused by ADAR1. Thus, these data suggest that ADAR1 promotes METTL3 expression in breast cancer cell lines.

### 2.3. ADAR1 Promotes the Progression of Breast Cancer through METTL3

To investigate whether ADAR1 plays an important role through METTL3 in MCF-7 and MDA-MB-231 cells, CCK-8 and colony formation assays were conducted. The results showed that cell proliferation co-transfected with ADAR1-overexpressing plasmid and siRNA against METTL3 was significantly inhibited compared to that of cells co-transfected with ADAR1-overexpressing plasmid and control siRNA (Figure 3a–c and Appendix A). The cell migration and invasion induced by ADAR1 were attenuated in the cells co-transfected with ADAR1-overexpressing plasmid and siRNA against METTL3 compared to that of cells co-transfected with ADAR1-overexpressing plasmid and control siRNA (Figure 3d–g). Moreover, we found that the cell proliferation, migration and invasion of MCF-7 and MDA-MB-231 cells co-transfected with METTL3-overexpressing plasmid and ADAR1 sgRNA were not significantly changed compared to that of cells co-transfected with METTL3-overexpressing plasmid and control sgRNA (Figure 3h–k and Appendix A). Taken together, these results indicate that ADAR1 promotes the progression of breast cancer through METTL3.

### 2.4. ADAR1 Intereacts with METTL3 mRNA in an RNA Editing-Dependent Manner and Alters METTL3 Binding Site of miR-532-5p

Next, we investigated the possible mechanism of how ADAR1 promotes METTL3 expression. We performed a RIP experiment by using ADAR1 antibody to detect the interaction between ADAR1 and METTL3 mRNA in breast cancer cell lines. As shown in Figure 4a, METTL3 mRNA was immunoprecipitated by ADAR1 antibody, and immunopreciated-METTL3 mRNA level was decreased after knockdown of METTL3 in both MCF-7 and MDA-MB-231 cell lines compared to the siCtrl control group. As ADAR1 is an RNA-editing enzyme, we wanted to see if the RNA editing activity of ADAR1 plays an important role in regulation of METTL3. We transfected plasmid expressing wild-type ADAR1-p150 protein and plasmid expressing mutant type ADAR1-p150 (▲ E/A) protein, an RNA-editing inactive form of ADAR1 in which glutamate was changed to alanine, into breast cancer cell lines, respectively, to detect METTL3 mRNA levels. We found that METTL3 mRNA level was significantly decreased in the ▲ E/A group compared to the wild-type ADAR1-p150 group (Figure 4b). mRNA m^6^A level was also decreased after transfecting ▲ E/A plasmid in the cells compared to the ADAR1-p150 wild-type group (Figure 4c). These results indicated that the RNA editing activity of ADAR1 is important for regulation of METTL3. We further performed DNA sequencing to detect whether A-to-I RNA editing of METTL3 mRNA by ADAR1 occurs in the cells transfected with plasmid expressing wild-type ADAR1-150 or mutated ADAR1-p150 protein. The results showed that A-to-I RNA editing of the nearby open reading frame (ORF) stop codon of METTL3 mRNA occurred (UCA(Ser)→UCG(Ser), same-sense mutation) in the wild-type ADAR1-p150 overexpression group, while A-to-I RNA editing of METTL3 mRNA was not observed in the ▲ E/A group (Figure 4d). This result further supported the hypothesis that the RNA editing activity of ADAR1 is essential for regulation of METTL3.

miRNAs play an important role in gene transcription and cell differentiation, proliferation and apoptosis [29]. Therefore, we investigated whether microRNAs play a vital role in breast cancer tumorigenesis by affecting METTL3 expression. Based on a series of analyses, including miRNA targeting analysis using RNA hybrid software and differential expression analysis, we identified a mature miRNA miR-532-5p that potentially regulates METTL3 by precisely binding to the sites of ADAR1–METTL3 interaction (Figure 4e). Moreover, miR-532-5p was downregulated in the cancer tissues compared to the adjacent tissues in the public database (Figure 4f). We further found that miR-532-5p mimics inhibited cell proliferation and colony formation and migration and invasion of breast cancer cell lines (Appendix A–d). Importantly, miR-532-5p mimics reduced METTL3 expression (Figure 4g,h). The METTL3 mRNA level was not significantly reduced after transfecting miR-532-5p mimics when we overexpressed ADAR1-p150 protein in the cells compared to the miR-NC group (Figure 4i). The results further suggested that ADAR1 edited METTL3 mRNA, which altered the binding site of miR-532-5p in METTL3 mRNA, leading to increased METTL3 mRNA and protein levels. Lastly, a luciferase report assay was performed to detect direct interactions between METTL3 mRNA and miR-532-5p in HEK293T cells transfected with firefly luciferase reporter, in which the 3/UTR of luciferase ORF contained the ADAR1 editing site (wild-type) or the edited site (mutant) of METTL3 mRNA. As shown in Figure 4j, the luciferase activity was greatly decreased in the wild-type group after transfecting miR532-5p mimics while not being changed in the mutant group. When we transfected cells with plasmid expressing ADAR1-p150 protein and miR-532-5p mimics or miR-NC, the cell proliferation, colony formation and cell migration and invasion did not show any differences between the miR-532-5p and miR-NC groups (Figure 4k–n). Together, these data suggest that ADAR1 promotes breast cancer tumor progression, likely by eliminating the inhibitory effect of miR-532-5p on METTL3 mRNA.

### 2.5. ADAR1 Upregulates ARHGAP5 Expression through METTL3

To screen the downstream genes of ADAR1 and METTL3, we performed Pearson correlation analyses separately for three independent datasets to identify genes correlated with both ADAR1 and METTL3 (Benjamini–Hochberg adjusted *p* < 0.05 and Pearson correlation r > 0.4). Seven genes, such as ARHGAP5, were revealed to be common to the three datasets (Figure 5a). First, the three independent datasets illustrated that ARHGAP5 expression was positively correlated with ADAR1 and METTL3 (Figure 5b,c). Second, Kaplan–Meier analysis indicated that higher ARHGAP5 expression levels are correlated with shorter overall survival (Figure 5d). Third, we found using IHC that ARHGAP5 expression was increased in breast cancer tissues compared to normal adjacent tissues (Figure 5e). Finally, knockdown of ARHGAP5 inhibited cell proliferation, colony formation and cell migration and invasion (Appendix A–d). These data imply that ARHGAP5 could promote breast cancer tumorigenesis and cancer development. qRT-PCR analysis showed that ADAR1-p150 or METTL3 overexpression enhanced ARHGAP5 mRNA expression, whereas ADAR1 or METTL3 silencing inhibited ARHGAP5 mRNA expression (Figure 5f,g), indicating that ADAR1 and METTL3 could promote ARHGAP5 expression. In addition, the RIP assay using an anti-METTL3 antibody and qRT-PCR further illustrated that ADAR1-p150 and METTL3 overexpression increased the amount of m^6^A-modified ARHGAP5 mRNA (Figure 5h). To determine whether ADAR1 overexpression enhances ARHGAP5 expression through METTL3, we knocked down METTL3 and overexpressed ADAR1-p150 protein and knocked down ADAR1 and overexpressed METTL3 in breast cancer cell lines to detect ARHGAP5 expression level by qRT-PCR. As shown in Figure 5i, ARHGAP5 level was decreased in METTL3-silenced and ADAR1-overexpressed cells but not changed in ADAR1-knockdown and METTL3-overexpressed cells compared to the corresponding control groups. Further, m^6^A antibody was used to detect and confirm methylated mRNA by using RIP-Seq and RIP, respectively. The results showed that ARHGAP5 was methylated by METTL3 (Figure 5j–l). These data show that ADAR1 upregulates ARHGAP5 through METTL3.

### 2.6. The Reader Protein YTHDF1 Increases ARHGAP5

To investigate the biological roles of ADAR1/METTL3/ARHGAP5 in breast cancer cells, we silenced ARHGAP5 and overexpressed ADAR1-p150 or METTL3 proteins in the cell lines, and then performed CCK-8, colony formation and transwell assays to detect cell proliferation, migration and invasion. We found that cell proliferation, colony formation and cell migration and invasion were significantly inhibited in the cells with knockdown of ARHGAP5 and overexpression of ADAR1-p150 (Figure 6a,c,e,f) or METTL3 proteins (Figure 6b,d,g,h and Appendix A) compared to the corresponding control group. These results further demonstrate that ADAR1 promotes breast cancer progression through the METTL3/ARHGAP5 axis.

Modification of mRNA by m^6^A is recognized by the corresponding proteins named “m^6^A reader proteins”, which play an important role in tumorigenesis. m^6^A reader proteins usually contain a YT521-B (YTH)-like domain, such as YTHDF1, YTHDF2 or YTHDF3 [31], and recognize and directly bind to mRNA during m^6^A methylation to modulate mRNA stability and/or translation [32]. It has been reported that YTHDF1 is elevated in breast cancer samples and promotes breast cancer progression [33]. We also observed that YTHDF1 was increased in breast cancer samples (Figure 6i and Appendix A). Therefore, we silenced YTHDF1 in breast cancer cell lines to investigate whether YTHDF1 has effects on ARHGAP5 mRNA and protein levels. As shown in Figure 6j, ARHGAP5 protein level was decreased after silencing YTHDF1 in both MCF-7 and MDA-MB-231 cells without affecting mRNA level. Next, we performed RIP experiments using YTHDF1 antibody followed by qRT-PCR to determine whether YTHDF1 can bind ARHGAP5 mRNA. The results showed that YTHDF1 binds ARHGAP5 mRNA (Figure 6k). We further performed a ribosome immunoprecipitation assay using Flag antibody to detect ARHGAP5 mRNA ribosome occupancy mediated by its m^6^A modification in breast cancer cells transfected with Flag-tagged RPL22 (ribosomal protein L22) upon YTHDF1 silencing [34,35]. As shown in Figure 6l, ribosomes accumulated ARHGAP5 mRNA in siCtrl cells, while ARHGAP5 mRNA ribosome occupancy was reduced in YTHDF1-silenced cells.

We investigated whether the expression level of ARHGAP5 is correlated with mRNA stability upon YTHDF1 silencing. To this end, we performed qRT-PCR to measure ARHGAP5 mRNA levels after transcription inhibition with actinomycin D. We showed that ARHGAP5 mRNA stability was not changed upon YTHDF1 knockdown (Figure 6m). Finally, we silenced YTHDF1 and treated the cells with or without MG132 and then measured ARHGAP5 protein level to explore whether protein decay can play a role in protein stability. We showed that YTHDF1 can modulate ARHGAP5 protein without affecting protein degradation (Figure 6n). Moreover, we found that knockdown of YTHDF1 inhibited cell proliferation, colony formation, migration and invasion in breast cancer cell lines (Appendix A–d). All above results indicate that YHTDF1 is a reader protein of ARHGAP5 m^6^A in breast cancer cell lines and that METTL3/YTHDF1 controls ARHGAP5 protein levels with the translation modulation mechanism as the post-transcriptional axis.

### 2.7. Loss of ADAR1 Suppresses Breast Cancer Growth and Decreases METTL3 and ARHGAP5 Expression In Vivo

It has been reported that loss of ADAR1 inhibits breast cancer growth in vivo. We confirmed this in nude mice by injecting ADAR1-loss breast cancer cell lines into the breast fat pad (Figure 7a–c,g–i). We further found that METTL3 and ARHGAP5 mRNA and protein levels were significantly decreased in ADAR1-loss cancer tissues compared to the Ctrl sgRNA group (Figure 7d–f). Interestingly, we did not observe obvious tumor growth in nude mice injected with ADAR1-loss MDA-MB-231 cells (Figure 7g–i). This finding is consistent with a previous study that showed that cancer growth is ADAR1-dependent in triple-negative breast cancer [36].

## 3. Discussion

In this study, we found that ADAR1, an A-to-I RNA-editing enzyme, interacts with METTL3 mRNA and increases its protein level to promote the proliferation, migration and invasion of breast cancer cells through a mechanism connecting ADAR1, METTL3 and YTHDF1. We showed that overexpression and knockdown of ADAR1 increases and decreases METTL3 mRNA and protein levels in breast cancer cell lines, respectively, which indicates that ADAR1 positively correlates with METTL3. Moreover, ADAR1 edits the nearby ORF stop codon of METTL3 mRNA, leading to its binding site changing to miR-532-5p and resulting in increased METTL3 protein, which further targets ARHGAP5 to promote breast cancer progression in a YTHDF1-dependent manner. Additionally, loss of ADAR1 inhibits breast cancer growth and decreases METTL3 and ARHGAP5 in vivo. Our findings identify the ADAR1–METTL3 axis as a novel important pathway that connects A-to-I editing and m^6^A RNA modifications in breast cancer progression.

We observed that both ADAR1 and METTL3 are upregulated in breast cancer samples compared to normal controls, and ADAR1 positively correlated with METTL3 in TCGA data and in our study. However, whether and how ADAR1 interacts with METTL3 in breast cancer has not been reported. A recent study has shown that ADAR1 is one of the targets of METTL3 and plays an oncogenic role in glioblastoma [35]. Another study demonstrated that m^6^A promotes A-to-I editing mediated by ADAR1 to inhibit aberrant innate immune responses against viruses [37]. In addition, Xiang, et al. reported that inhibition of m^6^A-catalyzing enzymes resulted in global A-to-I RNA-editing changes [38] thanks to a set of writer and eraser proteins [39]. All of theses studies indicate that ADAR1 activity or A-to-I editing mediated by ADAR1 can be affected by METTL3 or m^6^A to influence pathological processes depending on context. However, in our present study, we found that ADAR1 directly interacts with METTL3 mRNA, as evidenced by RIP using ADAR1 antibody, and that ADAR1 edits the nearby stop codon of METTL3 mRNA and changes its miR-532-5p binding site, leading to increased METTL3 mRNA and protein levels in breast cancer cells.

miRNAs perform biological functions by inducing degeneration of mRNA targets or blocking translation to prevent gene expression [40]. In this study, ADAR1-mediated A-to-I editing of the nearby ORF stop codon of METTL3 mRNA changes its miR-532-5p binding site and further prevents miR-532-5p from binding to METTL3 mRNA, leading to an increase in METTL3 mRNA and protein levels. miR-532-5p has been reported to play a vital role in the occurrence and development of many cancers, such as glioma cancer [41], lung cancer [42] and breast cancer [43]. It has also been reported that several microRNAs, including miR-186, miR-4429, miR-600 and let-7g, inhibit METTL3 by targeting METTL3 mRNA [42,43,44,45], while miR-24-2 promotes METTL3 transcription [46]. However, in our study, we found that miR-532-5p is a tumor suppressor and targets METTL3 mRNA in breast cancer cells. Therefore, we suggest that miR-532-5p promotes METTL3 expression by no longer binding to its edited site through ADAR1.

m^6^A modification is catalyzed by the methyltransferase complex (MTC), in which METTL3 is the core catalytic subunit. Studies have revealed that METTL3 plays vital roles in a variety of cancer types, either as an oncogene promoting the initiation and development of cancers [25,30,44,45,46] or as a tumor suppressor in some cases [47,48], via different mechanisms depending on the specific type of cancer. In this study, we found that METTL3, as an oncogene, is upregulated by ADAR1 in an RNA-editing dependent manner in breast cancer cell lines. Zhu et al. reported that METTL3 interacts with ARHGAP5 mRNA and increases its m^6^A modification, leading to chemoresistance to gastric cancer [49]. We further confirmed that ARHGAP5 is a target of METTL3 in breast cancer and promotes the proliferation, invasion and migration of breast cancer cells [50]. Finally, we found that METTL3-mediated m^6^A modification of ARHGAP5 mRNA promotes mRNA translation via YTHDF1, an m^6^A reader protein. It has been reported that mRNA m^6^A methylation is recognized by proteins containing YTH domains (YTHDFs) [31]. Chen et al. reported that YTHDF1 is overexpressed in both breast cancer cells and clinical breast cancer tissues [33]. We also found that YTHDF1 was increased in breast cancer samples. When we knocked down YTHDF1 using siRNAs in breast cancer cell lines, ARHGAP5 protein level was significantly decreased, without affecting its mRNA level. We further found that YTHDF1 interacts with ARHGAP5 mRNA and promotes its translation without affecting either its mRNA stability or protein decay. All the above results indicate that METTL3 upregulated by ADAR1 methylates ARHGAP5 to promote breast cancer progression in a YTHDF1-dependent manner, which enhances ARHGAP5 mRNA translation. However, details about how METTL3 regulates YTHDF1-recognized ARHGAP5 to promote breast cancer progression need to be investigated in the future. It should be noted that we did not examine the possibility that ADAR1 may impact ARHGAP5 mRNA or protein to affect tumor progression in this study, which will also be explored in the future.

It is well-known that ADAR1 plays an important role in tumorigenesis. Herein, we further confirmed that ADAR1 is a key protein in breast cancer. Loss of ADAR1 inhibits breast cancer growth and decreases METTL3 and ARHGAP5 expression levels in vivo.

In summary, our findings revealed the novel molecular pathway ADAR1/METTL3/ARHGAP5 connecting ADAR1, METTL3 and YTHDF1 (Figure 8), which may indicate ADAR1 or METTL3 as prognostic and therapeutical targets for the treatment of breast cancer.

## 4. Materials and Methods

### 4.1. Human Tissues, Cell Lines and Cell Culture

Human breast cancer tumors and corresponding adjacent, non-cancerous breast tissues (normal tissues) from 40 breast cancer patients, including 11 cases of stage I, 14 cases of stage II and 15 cases of stage III, were obtained from Sichuan Mianyang 404 hospital (Sichuan, China). The tissues were dissected and embedded in paraffin for immunohistochemistry analysis. All patients signed the patient informed consent form.

Human breast cancer cell lines MDA-MB-231, T47D, SK-BR-3 and MCF-7 were originally purchased from the American Type Culture Collection (ATCC, Manassas, VA, USA) and cultured in DMEM (Hyclone, Logan, UT, USA) supplemented with 10% fetal bovine serum (FBS, PAN-Biotech, Adenbach, Germany), 100 U/mL penicillin and 100 μg/mL streptomycin in a humidified air atmosphere of 5% CO_2_ at 37 °C.

### 4.2. Plasmids, miRNA Mimics, siRNAs and Transfection

The wild-type ADAR1 plasmids ADAR1-p150 and ADAR1-p110 and the mutant-type ADAR1 plasmid ADAR1-p150-∆E/A were obtained from Dr. Qingde Wang [51]. pcDNA3-Flag-METTL3 (plasmid #53739) was purchased from Addgene (Cambridge, MA, USA). The luciferase reporter plasmid with 3′UTR containing either the sequence of the METTL3 mRNA (1401–1681) edited by ADAR1 (called the METTL3 MT plasmid) or not edited by ADAR1 (METTL3 WT) was constructed by inserting it into the psichake2 vector between the Hind III and the XbaI sites. All of the constructs were verified by sequencing. The miR-532-5p mimics, siRNAs and negative control were synthesized by Genepharma Inc. (Shanghai, China). The miR-532-5p mimics and siRNAs sequences are listed in Appendix A. When cells were ∼60% confluent, cells were transfected with plasmids, miR-532-5p mimics or siRNAs in 24-well plates or 6 cm tissue culture dishes (BD Company, Franklin Lakes, NJ, USA) using Lipofectamine 2000 (Invitrogen Corporation, Carlsbad, CA, USA), according to the manufacturer’s instructions.

### 4.3. Cell Proliferation and Colony Formation Assays

Cell proliferation and colony formation assays were performed as described previously [16,52]. Briefly, for cell proliferation assay, 5 µL of Cell Counting Kit-8 reagent (CCK-8) (Bimake, Shanghai, China) was added to each well of the culture plates. A_450_ values were detected at 450 nm on a Microplate Reader (BIO-TEK, Rockville, MA, USA). For the colony formation assay, cells were seeded in 6-well plates at a density of 1000 cells per well in triplicate. Colony formation was observed under an optical microscope (Leica, Wetzlar, Germany) after 10 days.

### 4.4. Cell Migration and Invasion Assays

Cell migration and invasion assays were performed as described previously [52]. Briefly, 5 × 10^4^ cells were inoculated into the upper chamber, which was either uncoated or pre-coated with Matrigel (BD Company, Franklin Lakes, NJ, USA), and incubated with serum-free DMEM medium. Then, 600 µL medium containing 20% FBS was added to the lower chamber. After 36 h of incubation, the cells on the upper filter surface were removed using a cotton swab, while cells on the bottom side of the filter were fixed with methanol for 30 min and stained with 0.4% crystal violet (Amresco, Solon, OH, USA) for 5 min. Three fields were randomly selected for analysis with a microscope. The experiment was performed in triplicate.

### 4.5. RNA Extraction, Real-Time PCR

Total RNA was extracted from cells and mouse tissues using Trizol reagent (Invitrogen Corporation, Carlsbad, CA, USA) following the manufacturer’s instructions. First, 0.5 µg RNA was reverse transcribed in a total reaction volume of 10 µL with a Reverse Transcription Kit (Takara, Dalian, China) according to the manufacturer’s instructions. PCR was performed using SYBR Green Master Mix (Toyobo, Japan). The relative gene expression levels were calculated using the 2^−ΔΔCt^ rules. GAPDH was used as an endogenous control. The primers used in the RT-PCR are listed in Appendix A.

### 4.6. RNA m^6^A Dot Blot Assay

Different amounts of total RNAs were denatured by heating at 65 °C for 5 min and spotted on an Amersham Hybond-N + membrane (GE Healthcare, Piscataway, NJ, USA). After UV crosslinking, the membrane was washed and blocked, then incubated with anti-m^6^A antibody (1:1000, #56593s, Cell Signaling Technology, Inc., Danvers, MA, USA) overnight at 4 °C and subsequently incubated with the HRP-conjugated secondary antibody and visualized with the ECL system (Invitrogen Corporation, Carlsbad, CA, USA). The membrane stained with 0.02% methylene blue (MB) was used as the RNA loading control.

### 4.7. Western Blotting Analysis

All concrete steps were carried out as in previous studies [16,52,53]. Anti-METTL3, anti-YTHDF1 and anti-ARHGAP5 antibodies were purchased from Abcam; anti-ADAR1 and anti-GAPDH antibodies were obtained from Santa Cruz Biotech.

### 4.8. RNA Immunoprecipitation Assay (RIP)

Cells were washed twice with ice-cold PBS and then resuspended in Cell Lysis/IP Buffer (Thermo Scientific, MA, USA) supplemented with protease and RNase inhibitor. The lysates were centrifuged at 12,000× *g* for 15 min at 4 °C to remove cell debris after incubation on ice for 30 min at 4 °C. Next, an appropriate amount of the antibody was added into the supernatant and incubated overnight at 4 °C on a rotator. Subsequently, 30 µL of Protein A/G-Coated Agarose Beads (Santa Cruz Biotechnology, Santa Cruz, CA, USA) was added into the supernatant, which was incubated at 4 °C for 1 h, followed by three washes in Wash Buffer. Finally, 0.5 mL of Trizol was added to the tube, then RNA was extracted following the manufacturer’s instructions and subjected to RT-PCR analysis for quantification.

### 4.9. MeRIP and MeRIP-Seq

m^6^A RNA immunoprecipitation was carried out with the GenSeq^TM^ m^6^A RNA IP Kit (GenSeq Inc., Guangzhou, China) according to the manufacturer’s instructions. The input and the m^6^A IP samples were used for qRT-PCR and RNA-seq library generation with the NEBNext^®^ Ultra II Directional RNA Library.

### 4.10. Dual-Luciferase Reporter Assay

The assay was performed as described previously [52]. In brief, the cells were seeded in 24-well plates. After 24 h, cells were transfected with luciferase reporter plasmid (METTL3 WT or METTL3 MT, described in Section 2.2) and miR-532-5p mimics, along with pRL-TK to normalize the transfection efficiencies. At 48 h after transfection, cells were lysed for 15 min at room temperature, and relative luciferase activities were measured with the dual-luciferase reporter assay kit (Promega, Madison, WI, USA).

### 4.11. Immunohistochemistry (IHC)

Immunohistochemistry was performed as described previously [52]. Briefly, tissues were fixed in formalin and embedded in paraffin. After antigen retrieval and endogenous peroxidase were blocked, the paraffin sections were incubated with anti-ADAR1 antibody (1:100), anti-METTL3 antibody (1:100) and anti-ARHGAP5 antibody (1:100) in 5% goat serum solution overnight at 4 °C, respectively. Then the sections were incubated with a biotinylated secondary antibody, and an Avidin–Biotin Complex Detection Kit (Solarbio, Beijing, China) was used to detect antigen–antibody complexes. The sections were counter-stained with hematoxylin.

### 4.12. Actinomycin D Treatment for RNA Stability Assay

Cells transfected with siRNA against YTHDF1 were treated with 5 μg/mL actinomycin D (ActD, Sigma-Aldrich, St. Louis, MO, USA) for 10 h to inhibit new RNA synthesis. Then, the cells were harvested at the indicated time points to extract the total RNA for quantitative RT-PCR, as described above.

### 4.13. MG132 Treatment

Cells were incubated with 1.25 μM of MG132 (Sigma-Aldrich, St. Louis, MO, USA) for 24 h to inhibit proteasome. The cells were then harvested to isolate proteins for Western blotting analysis, as described above.

### 4.14. Transfected Stable Cell Lines

To generate ADAR1 loss cells, breast cancer cell lines (MCF-7 and MDA-MB-231) were stably transfected with ADAR1 CRISPR/Cas9 plasmid expressing single-guide ADAR1 and Cas-9 (ADAR1sgRNA) or control CRISPR/Cas9 plasmid expressing single-guide control RNA and Cas-9 (termed CtrlsgRNA). Then, cells were selected with 1 μg/mL puromycin and expanded. The above plasmids were kindly provided by Professor Zeng Tu at Chongqing Medical University.

### 4.15. In Vivo Experiments

Roughly 1 × 10^7^ of MCF-7 or MDA-MB-231 cells stably transfected with ADAR1 sgRNA or Ctrl sgRNA (as described above) were injected subcutaneously into the thighs of the female athymic nude mice (5 weeks old, 17–18 g, three mice per group). Tumor growth was measured by measuring the width (W) and length (L) with calipers, and the volume (V) of the tumor was figured using the criterion V = (W^2^ × L)/2. At 42 days after injection, the mice were euthanized and tumors were removed and weighed. The tumor samples were further analyzed with IHC and Western blot assay. The animal studies were performed according to the institutional ethics guidelines for animal experiments and approved by the Chongqing Medical University Animal Care and Use Committee.

### 4.16. Statistical Analyses

All experiments were performed in triplicate. The data are expressed as the means ± S.D. (standard deviation) and analyzed using the software GraphPad Prism 8.0 (GraphPad Software Inc., San Diego, CA, USA) and SPSS 16.0 (SPSS, Chicago, IL, USA). Comparisons of two groups were analyzed using two-tailed Student’s *t*-tests and ANOVA. The survival analysis was performed with Kaplan–Meier plots and the log-rank test. Correlations were analyzed with Pearson’s correlation test. *p*-value of <0.05 was considered to be statistically significant.

## Figures and Tables

**Figure 1 ijms-23-09656-f001:**
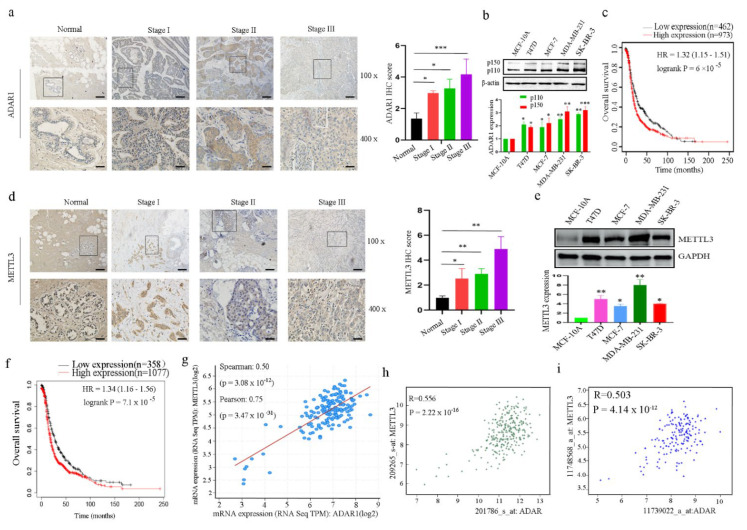
Both ADAR1 and METTL3 are upregulated in breast cancer tissues and cell lines. (**a**) Representative IHC staining micrographs of ADAR1 in breast cancer patients at different stages (magnification 100×, scale bar = 100 μm; magnification 400×, scale bar = 25  μm). Data are represented as means ± SD. * *p* < 0.05, *** *p* < 0.001. (**b**) ADAR1 protein levels assessed by Western blot were upregulated in breast cancer cell lines compared to normal human mammary epithelial cell line (MCF-10A). (**c**) Kaplan–Meier survival curves of ADAR1 in breast cancer patients (n = 1435). The median expression level was used as a cut-off to divide patients into low and high groups. (**d**) Representative IHC staining micrographs of METTL3 in breast cancer patients at different stages (magnification 100×, scale bar = 100 μm; magnification 400×, scale bar = 25 μm). Data are represented as means ± SD. * *p* < 0.05, ** *p* < 0.01. (**e**) METTL3 protein levels assessed by Western blot were upregulated in breast cancer cell lines compared to MCF-10A. (**f**) Kaplan–Meier survival curves of METTL3 in breast cancer patients (n = 1435). (**g**–**i**) The mRNA expression correlation of ADAR1 with METTL3 in three independent datasets.

**Figure 2 ijms-23-09656-f002:**
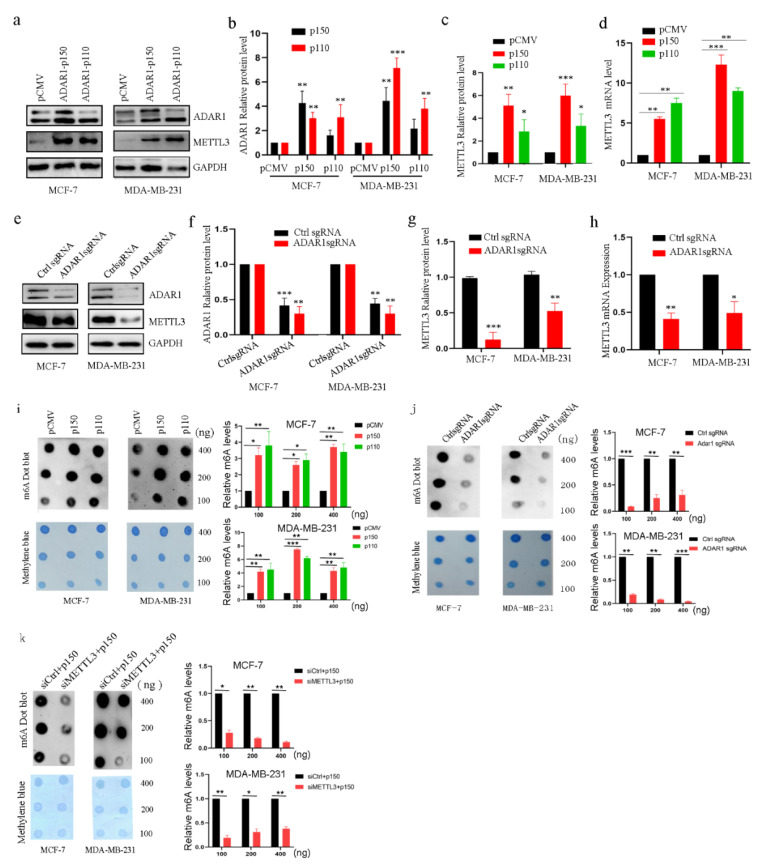
ADAR1 upregulates protein and mRNA levels of METTL3 and total mRNA m^6^A levels in breast cancer cells. (**a**–**d**) Overexpression of ADAR1 increased METTL3 protein and mRNA levels in breast cancer cell lines. Data are represented as means ± SD. * *p* < 0.05, ** *p* < 0.01, *** *p* < 0.001. (**e**–**h**) Depletion of ADAR1 decreased METTL3 protein and mRNA levels in breast cancer cell lines. Data are represented as means ± SD. * *p* < 0.05, ** *p* < 0.01, *** *p* < 0.001. (**i**) Overexpression of ADAR1 increased mRNA m^6^A levels, as detected by m^6^A dot blot in breast cancer cell lines. Methylene blue was used as an RNA loading control. Data are represented as means ± SD. * *p* < 0.05, ** *p* < 0.01, *** *p* < 0.001. (**j**) Depletion of ADAR1 decreased mRNA m^6^A levels, as detected by m^6^A dot blot in breast cancer cell lines. Methylene blue was used as an RNA loading control. Data are represented as means ± SD. ** *p* < 0.01, *** *p* < 0.001. (**k**) Knockdown of METTL3 decreased heightened mRNA m^6^A levels caused by ADAR1. Methylene blue was used as an RNA loading control. Data are represented as means ± SD. * *p* < 0.05, ** *p* < 0.01.

**Figure 3 ijms-23-09656-f003:**
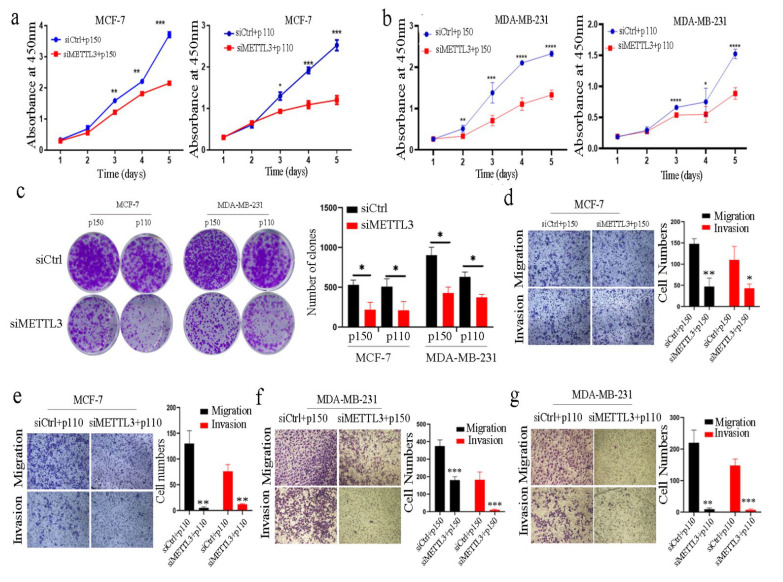
ADAR1 promotes breast cancer cell proliferation, migration and invasion mainly through METTL3. (**a**–**c**) CCK-8 and colony formation assays for cell growth and proliferation of breast cancer cells co-transfected with ADAR1-expressing plasmid and siRNA against METTL3 (c, scanned, 1×). Data are represented as means ± SD. * *p* < 0.05, ** *p* < 0.01, *** *p* < 0.001, **** *p* < 0.0001. (**d**–**g**) Cell migration and invasion were assayed with a transwell assay in breast cancer cells co-transfected with ADAR1-expressing plasmid and siRNA against METTL3 (magnification 100×). Data are represented as means ± SD. * *p* < 0.05, ** *p* < 0.01, *** *p* < 0.001. (**h**,**i**) CCK-8 and colony formation assays for cell growth and proliferation of breast cancer cells co-transfected with plasmid expressing METTL3 and ADAR1 sgRNA (i, scanned, 1×). (**j**,**k**) Cell migration and invasion were assayed with a transwell assay in breast cancer cells co-transfected with plasmid expressing METTL3 and ADAR1 sgRNA (magnification 100×). Data are represented as means ± SD.

**Figure 4 ijms-23-09656-f004:**
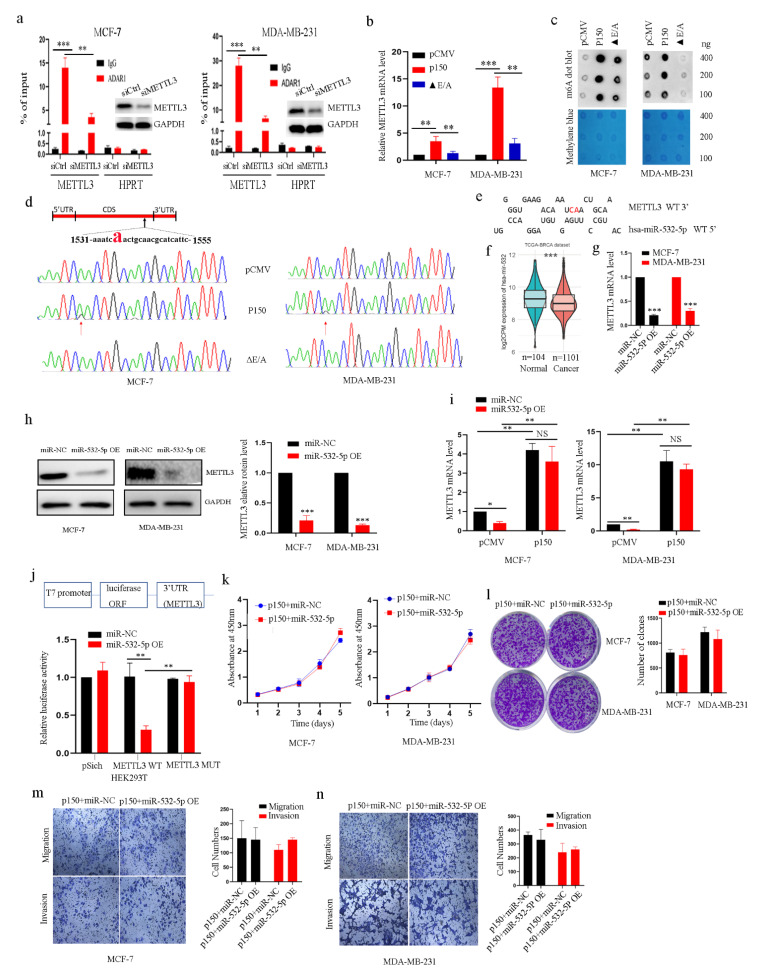
ADAR1 interreacts with METTL3 mRNA in an RNA-editing-dependent manner and alters its miR-532-5p binding site to promote breast cancer cell proliferation, migration and invasion. (**a**) RIP was performed to detect interactions between ADAR1 protein and METTL3 mRNA by using ADAR1 antibody in breast cancer cell lines transfected with siRNA against METTL3. Hypoxanthine guanine phosphoribosyl transferase (HPRT) served as a negative control. Data are represented as means ± SD. ** *p* < 0.01, *** *p* < 0.001. (**b**) METTL3 mRNA level was detected by RT-PCR in breast cancer cell lines transfected with plasmid expressing wild-type ADAR1-p150 protein or mutant ADAR1-p150 protein (▲ E/A). Data are represented as means ± SD. ** *p* < 0.01, *** *p* < 0.001. (**c**) mRNA m^6^A level was detected by m^6^A dot blot in breast cancer cell lines transfected with plasmid expressing wild-type ADAR1-p150 protein or mutant ADAR1-p150 protein. Methylene blue was used as an RNA loading control. (**d**) Sequencing of the RT-PCR products of METTL3 mRNA of breast cancer cell lines transfected with ADAR1-p150 or ADAR1-p150▲ E/A plasmid. Red letter “a “indicates editing site. Arrow indicates position of A-to-I editing. (**e**) The miR-532-5p binding sites in the region nearby the stop codon of METTL3mRNA. Red letter “a “indicates editing site. (**f**) Comparison of the expression of miR-532-5p between tumor and normal samples in breast cancer (n = 1205) based on the TCGA database. *** *p* < 0.001. (**g**,**h**) Quantitative analysis of METTL3 by qRT-PCR (**g**) and Western blot (**h**) in breast cancer cell lines transfected with miR-532-5p mimics. Data are represented as means ± SD. *** *p* < 0.001. (**i**) Quantitative analysis of METTL3 by qRT-PCR in breast cancer cell lines transfected with miR-532-5p mimics and plasmid expressing ADAR1-p150. Data are represented as means ± SD. * *p* < 0.05, ** *p* < 0.01. (**j**) The psiCHECK-METTL3-WT or psiCHECK-METTL3-MUT reporter construct was co-transfected with control or miR-532 mimics into HEK-293T cells. The dual-luciferase activities were determined at 48 h after transfection. Data are represented as means ± SD. ** *p* < 0.01. (**k**,**l**) CCK-8 and colony formation analysis for cell growth and proliferation of breast cancer cell lines transfected with plasmid expressing ADAR1-p150 and miR-532-5p mimics (l, scanned, 1×). (**m**,**n**) Cell migration and invasion were measured with a transwell assay in breast cancer cell lines transfected with plasmid expressing ADAR1-p150 and miR-532-5p mimics (magnification 100×). Data are represented as means ± SD. * *p* < 0.05, ** *p* < 0.01, *** *p* < 0.001.

**Figure 5 ijms-23-09656-f005:**
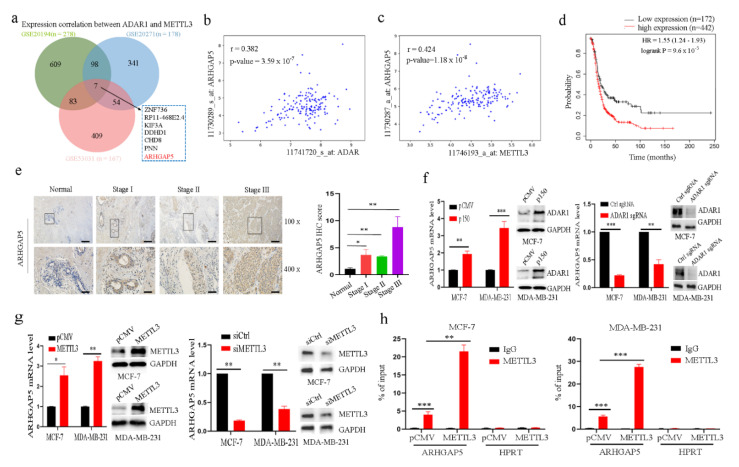
ADAR1 upregulates ARHGAP5 expression through METTL3. (**a**) The genes significantly correlated with both ADAR1 and METTL3 based on three independent datasets. (**b**,**c**) mRNA expression correlation of ADAR1 (**b**) or METTL3 (**c**) with ARHGAP5 in three independent datasets. (**d**) Kaplan–Meier survival plot for ARHGAP5 in breast cancer patients (n = 614). Median expression was used as a cut-off. (**e**) Representative IHC staining micrographs of ARHGAP5 in breast cancer tissues at different stages (magnification 100×, scale bar = 100 μm; magnification  400×, scale bar = 25 μm). * *p* < 0.05, ** *p* < 0.01. (**f**,**g**) Quantitative analysis of ARHGAP5 by RT-PCR in breast cancer cell lines transfected with plasmid expressing ADAR1 or METTL3 protein or plasmid silencing METTL3 or depleting ADAR1. Data are represented as means ± SD. * *p* < 0.05, ** *p* < 0.01, *** *p* < 0.001. (**h**) The interaction between METTL3 and ARHGAP5 mRNA was analyzed by RIP with METTL3 antibody in breast cancer cell lines transfected with plasmid expressing METTL3 protein. HPRT served as a negative control. ** *p* < 0.01, *** *p* < 0.001. (**i**) Quantitative analysis of ARHGAP5 mRNA with RT-PCR in breast cancer cell lines co-transfected with siRNA against METTL3 and plasmid expressing ADAR1 proteins or co-transfected with ADAR1 sgRNA and plasmid expressing METTL3 protein. Data are represented as means ± SD. ** *p* < 0.01, *** *p* < 0.001. (**j**) m^6^A-seq showed that ARHGAP5 was methylated by METTL3 in MDA-MB-231 cells silencing METTL3. (**k**) qRT-PCR of ARHGAP5 mRNA in MCF-7 and MDA-MB-231 cells after m^6^A immunoprecipitation. HPRT was used as negative control. Data are represented as means ± SD. *** *p* < 0.001. (**l**) m^6^A immunoprecipitation of ARHGAP5 mRNA upon METTL3 knockdown. Data are represented as means ± SD. ** *p* < 0.01, *** *p* < 0.001.

**Figure 6 ijms-23-09656-f006:**
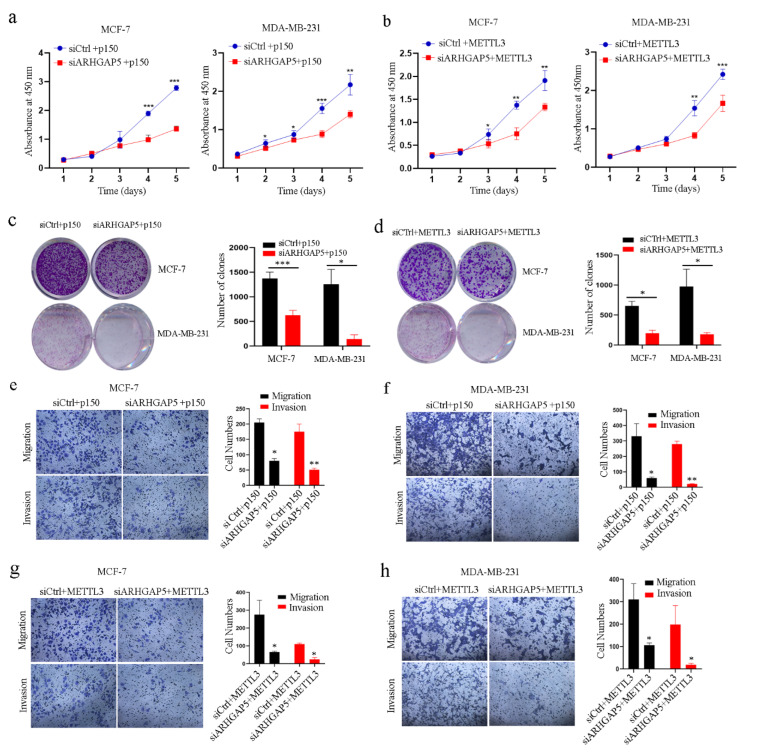
METTL3 promotes ARHGAP5 expression through the m^6^A reader protein YTHDF1. (**a**–**d**) CCK-8 and colony formation analyses for cell growth and proliferation in breast cancer cells co-transfected with siRNA against ARHGAP5 and plasmids expressing ADAR1-p150 protein (**a**,**c**) or with siRNA against ARHGAP5 and plasmids expressing METTL3 protein (**b**,**d**). c, d, scanned, 1×.Data are represented as means ± SD. * *p* < 0.05, ** *p* < 0.01, *** *p* < 0.001. (**e**–**h**) Cell migration and invasion were detected with a transwell assay in breast cancer cell lines co-transfected with siRNA against ARHGAP5 and plasmids expressing ADAR1-p150 protein (**e**,**f**) or with siRNA against ARHGAP5 and plasmids expressing METTL3 protein (**g**,**h**). (magnification 100×).Data are represented as means ± SD. * *p* < 0.05, ** *p* < 0.01. (**i**) Box plot showing YTHDF1 mRNA expression levels in breast cancer (red box) versus normal breast samples (gray box) (GEPIA (cancer-pku.cn)). ** *p* < 0.01. (**j**) qRT-PCR results for YTHDF1 and ARHGAP5 in MCF-7 and MDA-MB-231 breast cancer cells silencing YTHDF1. On the right, Western blotting of YTHDF1 and ARHGAP5 in siYTHDF1 cells. GAPDH was used as loading control. Data are represented as means ± SD. ** *p* < 0.01. (**k**) Relative enrichment of ARHGAP5 mRNA in YTHDF1-RIP in MCF-7 and MDA-MB-231cells. HPRT was used as negative control. Data are represented as means ± SD. * *p* < 0.05, ** *p* < 0.01. (**l**) Relative enrichment of ARHGAP5 mRNA in Flag-RIP in siYTHDF1 MCF-7 and siYTHDF1 MDA-MB-231 cells transfected with an RPL22-FLAG plasmid. Western blotting analysis and qRT-PCR results are shown for each cell. Data are represented as means ± SD. ** *p* < 0.01. (**m**) ARHGAP5 mRNA stability in the siYTHDF1 MCF-7 and MDA-MB-231 cells was determined by qRT-PCR after actinomycin D treatment at the indicated time points. On the right, the qRT-PCR results for YTHDF1 silencing. Data are represented as means ± SD. ** *p* < 0.01. (**n**) Western blotting of ARHGAP5 in siYTHDF1 and siCtrl MCF-7 and MDA-MB-231 breast cancer cells treated with or without 1.25 μM MG132 for 24 h. GAPDH was used as loading control. Data are represented as means ± SD. * *p* < 0.05.

**Figure 7 ijms-23-09656-f007:**
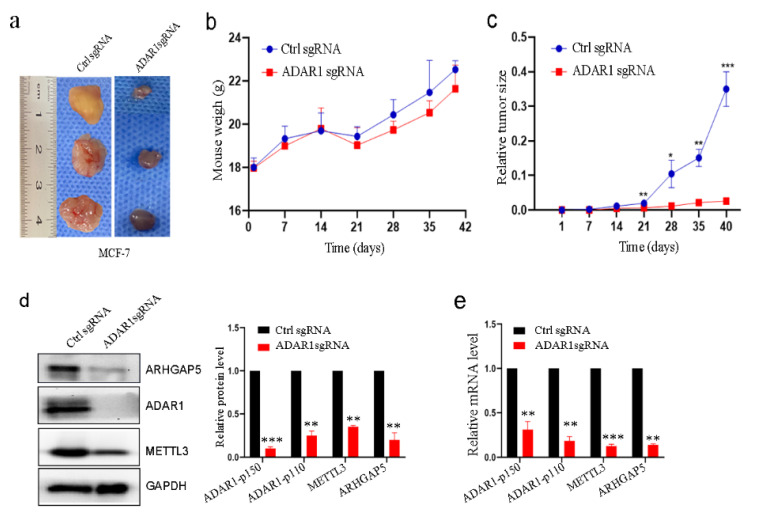
Loss of ADAR1 suppresses breast cancer tumor growth and decreases METTL3 and ARHGAP5 levels in vivo. (**a**) Xenografted MCF-7 cell tumors with stably expressing single-guide ADAR1(ADAR1 sgRNA) or single-guide control (Ctrl sgRNA) in nude mice. (**b**,**c**) Weight and tumor size were measured in xenografted MCF-7-cell nude mice. Data are represented as means ± SD. * *p* < 0.05, ** *p* < 0.01, *** *p* < 0.001. (**d**–**f**) Western blots, RT-PCR and IHC staining of ADAR1, METTL3 and ARHGAP5 in xenografted MCF-7 cell tumors (magnification 100×, scale bar = 100 μm; magnification 400×, scale bar = 25 μm). Data are represented as means ± SD. ** *p* < 0.01, *** *p* < 0.001. (**g**) Xenografted MDA-MB-231 cell tumors with stably expressing single ADAR1 sgRNA or Ctrl sgRNA in nude mice. (**h**,**i**) Weight and tumor size were measured in xenografted MDA-MB-231-cell nude mice. Data are represented as means ± SD, n = 3, *** *p* < 0.001.

**Figure 8 ijms-23-09656-f008:**
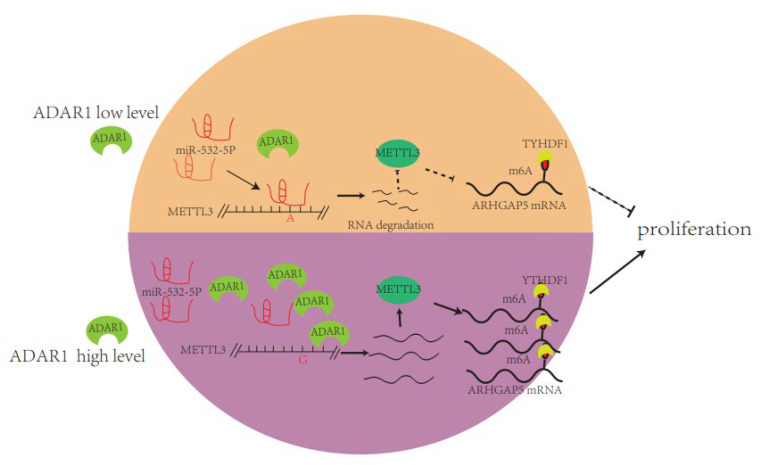
The molecular model of the ADAR1/METTL3/ARHGAP5/YTHDF1 axis promoting progression of breast cancer. The high level of ADAR1 in breast cancer edits METTL3 mRNA and changes its miR-532-5p binding site, leading to elevated METTL3, which, in turn, further methylates YTHDF1-recognized ARHGAP5 to promote the cell proliferation.

## Data Availability

Not applicable.

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
