# Peer review of "RNA Editing Enzyme ADAR1 Regulates METTL3 in an Editing Dependent Manner to Promote Breast Cancer Progression via METTL3/ARHGAP5/YTHDF1 Axis"

_ijms, 2022, doi:10.3390/ijms23179656_

Round 1

Reviewer 1 Report

In this manuscript, the authors provide evidence that ADAR1 regulates METTL3 expression in an editing-dependent manner and that regulation along with YTHDF1 regulation of ARHGAP5 levels result in increased breast cancer progression. Overall, the experiments are well performed and controlled. However, there are a few major points about the experimental outcomes that should be addressed as well as minor improvements to the text regarding citations and typographical/English grammar errors.

Major suggestion:

If ADAR1 promotes cancer progression via upregulation of METTL3, which in turn controls ARHGAP5, it is unclear why experiments such as those in Figure 5i – where METTL3 is upregulated and ADAR1 is downregulated- do not impact ARHGAP5 levels. This suggests that ADAR1 is also acting on ARHGAP5. Furthermore, as these same experiments (loss of ADAR1 in cells expressing METTL3) do not alter colony formation, proliferation, invasion or migration (Figure 3h-k), it is likely that ADAR1 has impacts on ARHGAP5 that are critical for tumor progression. The authors should examine this possibility or at least discuss this model and whether ARHGAP5 is known to be edited.

A few minor suggestions:

1.    The introduction does not have a reference until line 40. The authors should include references to recent reviews at the end of each of the introduction sentences prior to line 40. For example, general RNA modifications affecting biological processes (sentence 1): Wilkinson and He, Front Cell Dev Biol, 2022. Sentence 2 – general A-to-I editing and m6A reviews for cancer biology, ex. Baker AR, Slack Trends Genetics, 2022 (A-to-I editing) and Chen Z et al, Front Pharmacol, 2022 (m6A). Sentence 3 – general ADAR biology, ex Erdmann et al, Critical Reviews in Biochemistry and Molecular Biology, 2021, etc.

2.    Line 39, interferon (IFN) should not be plural

3.    Line 93 should include references.

4.    The sentence on lines 100-101 needs edited (methy should be methyl, etc).

5.    The sentence starting on line 124 needs edited.

6.    Line 162 should say mutated (currently mutanted).

7.    Lines 182-183 need edited.

8.    It would be easier for the reader if the samples in Figures 4f and 6i were reversed (i.e. Normal first then tumor values).

Reviewer 2 Report

In the manuscript “RNA Editing Enzyme ADAR1 Regulates METTL3 in an Editing 2 Dependent Manner to Promote Breast Cancer Progression Via 3 METTL3/ARHGAP5/YTHDF1 Axis”, Li et al explore the potential regulatory roles of ADAR1-METTL3 axis in breast cancer. The authors found that ADAR1 and METTL3 are upregulated in both breast cancer and cell lines, and the expression level of ADAR1 and METTL3 is correlated with poor prognosis. ADAR1 could upregulate the METTL3 gene, along with an increase in total mRNA m6A levels. Further investigation demonstrated that ADAR1 promotes the progression of breast cancer by regulating METTL3. The mechanisms research found that ADAR1 interacts with METTL3 mRNA in an RNA editing-dependent manner and alters the METTL3 binding site of miR-532-5p. Moreover, ADAR1 upregulates ARHGAP5 expression through METTL3, and the reader protein “YTHDF1” increases ARHGAP5. More importantly, the author’s work demonstrated that loss of ADAR1 suppresses breast cancer growth and decreases METTL3 and ARHGAP5 expression in vivo.  This topic is interesting, and this project was well-designed. There are several comments to help the authors improve their manuscript before release to publish.

Major

1. The authors explore the expression of ADAR1 and METTL3 in human breast cancer patient cohorts. More detailed information of the clinical cohorts should be added to this manuscript. For example, how many patient samples were analyzed in the current work? How do the authors get the normal tissues from humans? 

2. In Figures 1A and 1d, lack of scale bar. In those panels, the authors analyzed the expression of ADAR1 and METTL3 in different stages of breast cancer, including normal, early, medium, and late stages. How many samples of each stage were used? Additional statistical analysis will enhance the persuasiveness of the data.

3. In Figures 2i, 2j and 2k, the m6A levels should be quantified.

4. The authors used the HPRT as a control. What is HPRT? Why did the authors choose HPRT as a control? A brief introduction to HPRT should be added.

5. In Figure 5E, lack of scale bar. How many samples of each stage were used? Additional statistical analysis will enhance the persuasiveness of the data.

6. In Figure 7A, lack of scale bar. The size of the tumor in the middle position from the control group seems to have similar size to that of the last tumor from the experimental group. But in Figure 3A, the statistical data showed there is a huge difference between the control and experimental groups. The authors should check their data carefully.

7. In Figure 7F, where is the scale bar? The scale bar should describe in the figure legends.

8. In Figure 7G, lack of scale bar. In this figure, the western blot bands were overcutting.

9. The statistical analyses should be described in detail. For example, which method and software are used to analyze the difference between the experimental and control group? How many biological repeats were conducted?

10. The ethical statement of the human patient specimens and mice experiments should be added.  The authors should provide sufficient material to prove their work fully comply with the ethical requirements of the journal.

Minor

1. In Figure 1C, Lack of ordinate identification

2. In line 87, west blot should be the western blot.

3. In line 109, Over expression should be overexpression

4. In line 146, ADAR1antibody should be ADAR1 antibody

5. In lines 183, and 196, interrection should be interaction, right?

6. In line 246. bothADAR1 should be both ADAR1
